# Value of 11C-Methionine PET Imaging in High-Grade Gliomas: A Narrative Review

**DOI:** 10.3390/cancers16183200

**Published:** 2024-09-20

**Authors:** Zsanett Debreczeni-Máté, Omar Freihat, Imre Törő, Mihály Simon, Árpád Kovács, David Sipos

**Affiliations:** 1Doctoral School of Health Sciences, Faculty of Health Sciences, University of Pécs, 7621 Pécs, Hungary; 2Department of Public Health, College of Health Science, Abu Dhabi University, Abu Dhabi P.O. Box 59911, United Arab Emirates; 3Department of Oncoradiology, Faculty of Medicine, University of Debrecen, 4032 Debrecen, Hungary; 4Department of Medical Imaging, Faculty of Health Sciences, University of Pécs, 7621 Pécs, Hungary; 5Dr. József Baka Diagnostic, Radiation Oncology, Research and Teaching Center, “Moritz Kaposi” Teaching Hospital, Guba Sándor Street 40, 7400 Kaposvár, Hungary

**Keywords:** 11C-MET, hybrid imaging, PET/CT, multimodal imaging, glioma, high-grade glioma

## Abstract

**Simple Summary:**

Patients with high-grade gliomas have a poor prognosis, with a median survival of only 11–18 months from the initial diagnosis after standard treatment. Surgery and chemoradiation are the first-line treatment strategies for malignant gliomas, but achieving durable local tumor control is a major challenge due to the pervasive infiltrative nature of gliomas, their growth rate, and their recurrence rate. Gadolinium-enhanced magnetic resonance imaging (MRI) is currently the primary imaging technique for high-grade gliomas; however, it has limitations, including difficulty in differentiating between tumor recurrence and treatment-induced changes, as well as challenges in accurately assessing total tumor volume due to the restricted sensitivity of gadolinium contrast. It is therefore worth looking for alternative imaging modalities that can overcome these limitations. Positron emission tomography (PET) is one of them. To this end, the identification of PET-based markers that can be used in the imaging of high-grade gliomas is of paramount importance.

**Abstract:**

11C-Methionine (MET) is a widely utilized amino acid tracer in positron emission tomography (PET) imaging of primary brain tumors. 11C-MET PET offers valuable insights for tumor classification, facilitates treatment planning, and aids in monitoring therapeutic response. Its tracer properties allow better delineation of the active tumor volume, even in regions that show no contrast enhancement on conventional magnetic resonance imaging (MRI). This review focuses on the role of MET-PET in brain glioma imaging. The introduction provides a brief clinical overview of the problems of high-grade and recurrent gliomas. It discusses glioma management, radiotherapy planning, and the difficulties of imaging after chemoradiotherapy (pseudoprogression or radionecrosis). The mechanism of MET-PET is described. Additionally, the review encompasses the application of MET-PET in the context of primary gliomas, addressing its diagnostic precision, utility in tumor classification, prognostic value, and role in guiding biopsy procedures and radiotherapy planning.

## 1. Introduction

Positron emission tomography, also called PET or a PET scan, is a type of nuclear imaging. This is a non-invasive diagnostic technique that qualitatively and quantitatively assesses the biodistribution of a target molecule in vivo, typically using short-lived positron emitters that are produced in a medical cyclotron and are typically attached to a target molecule. By combining substances that occur naturally in the human body, it is possible to follow selected metabolic pathways and cell functions with great precision. The most common tracer used in PET diagnostics is 18F-fluorodeoxyglucose (18-FDG), which is widely used in oncology, cardiology, neurology, and inflammation diagnostics [1,2]. However, because high glucose metabolism in the brain reduces the accuracy of tumor delineation, the value of FDG-PET in diagnostics and radiotherapy planning for central nervous system (CNS) tumors is very limited [1].

Unlike FDG, radiolabeled amino acids such as [11C-methyl]-l-methionine (MET), O-(2-[18F]-fluoroethyl)-l-tyrosine (FET), and 3,4-dihydroxy-6-[18F]-fluoro-l-phenylalanine (FDOPA) exhibit minimal uptake in normal brain tissue, enabling the clear visualization of brain tumors against a high tumor-to-background signal [2]. The diagnosis of brain tumors, especially high-grade gliomas, which are the most common and have a high mortality rate, is the most common application of PET/CT with 11C-MET. The main advantage of using this technique is that very little tracer is taken up by healthy tissue. However, the mechanism underlying the accumulation of 11C-MET in pathological lesions has not yet been fully elucidated. There are three explanations for the increased uptake of 11C-methionine by tumor cells. First, tumor cells take up MET via an L-type amino acid transporter system and a sodium-dependent transporter system. This active transport is dependent on cell proliferation and tumor malignancy. Second, passive diffusion of MET may occur due to damage to the blood–brain barrier in intracranial malignant lesions. Third, MET uptake in tissue is also influenced by tumor vascularization [1,2,3].

11C-MET can be used to assess the extent of intracranial malignancies for radiotherapy planning and, more importantly, to confirm tumor recurrence after surgery and radiotherapy when recurrence is not clear on MRI and radionecrosis is suspected [4]. To create a PET tracer, one of the stable carbon atoms in a methionine molecule is replaced by a carbon-11 isotope, which has a half-life of 20.4 min. Due to this short half-life, 11C-MET cannot be transported over long distances and must be synthesized on-site at a medical facility equipped with a cyclotron. The imaging procedure is typically conducted 10–20 min after intravenous administration of the tracer, and no significant adverse effects related to 11C-MET administration have been reported in the literature [4,5].

At many imaging centers, nuclear medicine images are combined with computed tomography (CT) scans or magnetic resonance imaging (MRI) scans to create special views [3]. This is called image fusion, or co-registration. Image fusion lets physicians combine information from two separate examinations into one picture. This leads to more accurate information and a more precise diagnosis. Both scans can be carried out at the same time with positron emission tomography/computed tomography (PET/CT) scanners [1,2,3,4,5]. 11C-MET imaging involves the quantification of amino acid uptake in tissues, leveraging the enhanced methionine transport in tumors due to increased protein synthesis and metabolic activity. The outcome profile typically includes high contrast between the tumor and normal brain tissues, enabling the precise delineation of glioma boundaries, the assessment of tumor grade, and the monitoring of recurrence or response to therapy [3,4,5,6].

The aim of this study is to evaluate the diagnostic utility and clinical applicability of 11C-MET PET in the management of high-grade gliomas. Specifically, the study seeks to elucidate the underlying mechanisms of 11C-MET uptake in glioma cells, to assess its effectiveness in tumor delineation and radiotherapy planning, and to compare its accuracy with traditional imaging modalities, such as FDG-PET and MRI, particularly in distinguishing between tumor recurrence and radionecrosis. Additionally, the study aims to explore the integration of 11C-MET PET with CT and MRI for enhanced diagnostic precision.

## 2. Gliomas

Gliomas are the most common type of tumor in the central nervous system, accounting for approximately 80% of all malignant primary tumors of the brain. This type of tumor, which arises from the glial cells of the brain, is highly malignant, is prone to relapse, and has an extremely high mortality rate. Gliomas can be classified into grades I to IV based on the level of malignancy [7]. High-grade gliomas are tumors that have been diagnosed as either grade III or grade IV and represent the most aggressive and invasive form of glioma. The classification of CNS tumors was, for a long time, based on histological findings. In recent years, however, molecular biomarkers have become increasingly important in providing additional and more definitive diagnostic information. The fifth edition of the *WHO Classification of Central Nervous System Tumors* (WHO CNS5), therefore, includes a number of clinicopathologically useful molecular variables that are important for the most accurate classification of CNS tumors [8].

The updated 2021 WHO classification categorizes adult high-grade gliomas (HGGs) into four subtypes: grade 3 oligodendroglioma (1p/19q codeleted, IDH mutant), grade 3 IDH-mutant astrocytoma, grade 4 IDH-mutant astrocytoma, and grade 4 IDH wild-type glioblastoma (GB) [9].

This advancement has significantly improved diagnosis and treatment, addressing the overlap in tumor growth characteristics seen in earlier histological classifications. Key molecular alterations, such as IDH, P53, ATRX, and 1p/19q, are crucial for diagnosis and prognosis. For example, IDH-mutant GBM patients typically survive 24–36 months, compared to 12–18 months for IDH wild-type GBM. These molecular insights allow for better classification, and gliomas with lower-grade histology but higher-grade molecular features are now treated as high-grade gliomas [9,10].

The intensity of 11C-MET uptake in a tumor is correlated with its grade and malignancy: tumors with higher proliferative potential and higher histological malignancy tend to show a greater accumulation of the tracer. Thus, amino acid PET adds valuable information for the differential diagnosis of CNS lesions suggestive of glial brain tumors, but neuropathological tissue evaluation remains mandatory in most patients for a definitive diagnosis [1,2,3,5,6,7].

A wide range of symptoms are experienced by glioma patients. Symptoms may be localized or generalized, depending on the size and location of the tumor and the degree of peritumoral edema. Unusual or frequent headaches, seizures, behavioral changes, difficulty with walking or balance, focal neurological deficits, and signs of increased intracranial pressure are common acute clinical features [7,11].

## 3. Imaging Techniques

### 3.1. Initial Diagnosis and Surgical Management

Most glioblastoma patients undergo a computed tomography (CT) scan of the brain upon initial presentation. After identifying a mass and ruling out hemorrhage, contrast-enhanced magnetic resonance imaging (MRI) is typically performed, incorporating standard sequences such as T2-weighted (T2w), T2 fluid-attenuated inversion recovery (T2-FLAIR), gradient echo, T1-weighted (T1w), and T1 contrast-enhanced (T1CE) imaging [12].

Maximum safe resection is recommended as the initial standard of care whenever possible. Neurosurgeons often use high-resolution MRI (0.5–1.2 mm slice thickness) for surgical planning and intra-operative guidance and to determine how aggressive the resection should be based on the risk of toxicity to nearby vital regions [12,13].

PET imaging is an important modality in neuro-oncology, not only for diagnosis but also for biopsy or surgical planning. Biopsies are performed for diagnosis in de novo cases, when the tumor location is irreversible and resection cannot be performed without compromising normal function. Biopsy planning is most commonly based on contrast-enhanced MRI information. However, given the frequent inter- and intratumoral heterogeneity, a modality that provides only morphological information often underestimates the extent of the tumor, as most high-grade gliomas do not show clustering on gadolinium-enhanced T1 images. In contrast, PET imaging provides the best definition of the biopsy site and tumor extent by determining the maximum activity of the lesion. However, a combination of the two modalities provides the highest sensitivity and specificity [11,14,15].

### 3.2. Post-Operative Imaging and Radiation Planning

Early post-operative MRI is performed primarily to determine the extent of resection and to assess residual disease. It is recommended to be performed within 48–72 h post-operatively due to the contrast enhancing the effects of surgery, such as inflammation or scarring. Radiotherapy is usually started 3–6 weeks after surgery. This allows for an adequate recovery period after surgery. The goal of radiotherapy is to improve local control without inducing neurotoxicity. The planning of radiotherapy involves the registration (also known as “fusion”) of the post-operative MRI (T1CE and T2-FLAIR sequences) with the planning simulation CT scan, which allows the delineation of T2-FLAIR abnormalities and residual enhancement during treatment planning [11,15].

PET imaging in radiotherapy has introduced the concept of biological target volume (BTV), which complements the traditionally used volumes: gross tumor volume (GTV), clinical target volume (CTV), and planning target volume (PTV) [7]. In addition to conventional brain tumor imaging modalities, considerable emphasis is placed on integrating positron emission tomography imaging, which provides insight into biological and functional aspects of tumor morphology. The main advantage of amino acid PET tracers and amino acid analog PET tracers is their ability to show a relatively high uptake in the tumor tissue while at the same time showing a comparatively low uptake in the normal brain tissue. This property allows for the precise delineation of active tumor volumes, even in regions where there is no contrast enhancement on conventional magnetic resonance imaging. For the more precise identification of active tumors, the use of multimodal imaging with 11C-MET-PET combined with MRI is promising. As a result, smaller clinical target volume margins may be required, allowing higher doses to be delivered to regions containing active tumor cells. This precision can enhance local control and overall survival in high-grade glioma patients. Consequently, many centers now incorporate amino acid imaging into CT- and MRI-based treatment planning, particularly for targeted radiotherapy, dose escalation, and recurrent disease management [14,15,16,17].

### 3.3. Difficulties with Post-Treatment Imaging

Following treatment for brain tumors, differentiating treatment-related changes from actual tumor recurrence remains a challenge and is of paramount clinical importance, with significant implications for clinical management. Pseudoprogression occurs when imaging scans show that the tumor appears to grow after treatment. The imaging features of most radiation necrosis (RN) are similar to those of malignant glioma on computed tomography or magnetic resonance imaging. This makes it difficult to differentiate between glioma recurrence and RN. However, what looks like tumor growth on the scans is actually a response to treatment. The misinterpretation of treatment-related changes as tumor progression can lead to unnecessary and premature discontinuation of an effective treatment option, with a potentially negative impact on survival. In addition, the efficacy of subsequent treatment may be overestimated, leading to misleading results in trials evaluating treatment options for recurrent disease. This is the most common clinical indication for amino acid PET, requested in almost 50% of glioma patients [17,18].

However, it should be noted that mild but increased amino acid tracer uptake may also occur—although much less frequently—in non-neoplastic lesions (e.g., acute or subacute cerebral ischemia, brain abscess, inflammatory lesions associated with active multiple sclerosis or status epilepticus). In addition, 20–30% of patients with WHO grade 2 CNS gliomas with an isocitrate dehydrogenase (IDH) gene mutation show no amino acid uptake [14,15,16,17,18].

## 4. Treatment

There are many ways to try to treat gliomas, but none is perfect. Treatment options depend on the tumor type, grade, size, and location. Modern treatments include surgery, chemotherapy, radiotherapy, and targeted drug therapy or a combination thereof. However, despite this comprehensive treatment regimen, patients experience tumor recurrence within a median time of 6–7 months [18,19,20,21].

### 4.1. Surgery

The primary goal of surgery is to confirm the diagnosis by determining whether the tumor is cancerous. Another key objective is to remove as much of the tumor as possible while preserving neurological function. Additionally, surgery can help alleviate intracranial pressure and manage symptoms like seizures that are difficult to control [20,21].

Determining the appropriate surgical approach is a complex decision made by the neurosurgeon in collaboration with a multidisciplinary team. Key factors considered include the patient’s age, Karnofsky Performance Status (KPS), and the tumor type, grade, and location, all of which influence the choice of surgery and the potential risks and benefits [20,21].

Resection is a major surgical procedure aimed at removing a significant portion of tissue. A gross total resection aims to remove all visible tumor tissue, offering a better prognosis, while a subtotal resection removes only part of the tumor. In cases where the glioma is located in a difficult-to-access or critical brain region, resection may not be feasible, and a biopsy is performed instead. There are two types of biopsies for gliomas: stereotactic biopsy, which is minimally invasive, and open biopsy, involving a more extensive surgical approach [20].

High-grade gliomas (HGGs) cannot be fully treated with surgery alone, as infiltrating glioma cells extend beyond the visible tumor mass and are not detectable on imaging. These residual cells remain after surgery and serve as a source for tumor recurrence [21,22].

### 4.2. Radiation Therapy

Radiation therapy targets tumor cells with high-energy rays, delivered locally, to destroy them by inducing non-specific breaks in the DNA strands of rapidly dividing cells.

Different types of radiotherapy are used to treat different types of glioma, as described below.

External beam radiation therapy (EBRT) is the most common method of treating gliomas. EBRT involves delivering tightly focused beams of radiation from outside the body. Daily treatments (fractions) are given over several weeks during a course of EBRT. The radiation machine (usually a linear accelerator) that produces and directs the radiation beams is controlled by the radiation oncology team. EBRT usually uses X-rays (also known as photons), but it can also use electrons or other less common particles, such as protons [23].

Three-Dimensional Conformal Radiation Therapy, or 3D-CRT, is an advanced technique that uses imaging technologies to create three-dimensional images of a patient’s tumor and nearby organs and tissues. This allows higher and more effective doses of radiation to reach the cancer cells directly. At the same time, it is possible to significantly reduce the amount of radiation that is received by the surrounding healthy tissue [24].

Intensity-Modulated Radiation Therapy (IMRT) is an advanced, high-precision radiotherapy technique that employs computer-controlled linear accelerators to deliver targeted radiation doses to malignant tumors or specific regions within them. By modulating the intensity of the radiation beams across multiple small volumes, IMRT enables precise dose distribution, closely conforming to the 3D shape of the tumor and minimizing exposure to surrounding healthy tissues. In addition, IMRT can focus more radiation on the tumor and minimize the dose to surrounding critical normal structures [25].

Stereotactic radiosurgery (SRS) is not surgery in the literal sense. It is a form of radiotherapy. It uses photon or proton beams to deliver a high dose of radiation to a small, precise area. This means that SRS can be given in fewer treatment sessions than other types of radiotherapy—usually between 1 and 5 sessions. SRS is often used for patients who cannot tolerate surgery or for tumors that are difficult to access surgically. Gamma Knife or CyberKnife are forms of SRS [26,27].

High-grade gliomas represent a challenging group of brain tumors due to their aggressive nature and infiltrative growth. Successful management of these tumors requires a multidisciplinary approach, with accurate and comprehensive treatment planning playing a key role. Medical imaging techniques such as PET-CT and MRI have revolutionized the assessment and management of high-grade gliomas. The integration of metabolic information from PET-CT with anatomical and functional data from MRI provides a comprehensive understanding of tumor biology and its relationship to surrounding structures. This hybrid approach helps to identify the areas of highest tumor activity, taking into account their proximity to vital brain regions [11,12,13,14,15,16,17].

Aggressive gliomas overexpress amino acid transporters, leading to the increased uptake of radiolabeled amino acids like 11C-MET, reflecting heightened tumor metabolism. This uptake enables the precise delineation of tumor extent, including infiltrative regions missed by other imaging modalities, and aids in tumor staging, treatment planning, and response monitoring, contributing to personalized treatment strategies and improved outcomes [11,12,13,14].

### 4.3. Chemotherapy

Chemotherapy uses drugs to target and destroy rapidly dividing cells, including cancer cells, by altering their DNA and disrupting growth, division, and survival. However, it can also harm healthy cells, leading to side effects. Glioma can be treated with a single chemotherapy drug or a combination of drugs. Temozolomide is the only chemotherapy drug used to treat glioma [28]. However, sometimes, a combination is chosen because they work better together. Procarbazine, lomustine, and vincristine (or PCV) are common combinations of drugs used to treat gliomas. Platinum-based drugs, such as cisplatin and carboplatin, may be used if the cancer returns after chemotherapy with temozolomide or PCV [29].

Chemotherapy is often combined with radiotherapy in various approaches. Concomitant treatment involves administering chemotherapy simultaneously with radiotherapy, while adjuvant treatment refers to giving chemotherapy after radiotherapy. In concurrent plus adjuvant treatment, chemotherapy is administered both during and after the course of radiotherapy.

### 4.4. Targeted Therapy

Targeted therapy drugs attack specific parts of tumor cells to slow their growth and spread. Targeted therapies only attack tumor cells and not healthy cells. This means they do not have the same side effects as standard chemotherapy. At the moment, there are only a small number of targeted therapies available for glioma. These are also only effective against gliomas that grow or spread through the specific enzymes, proteins, or other molecules that are being targeted [30,31].

### 4.5. Alternating Electric Field Therapy

Alternating electric field therapy, or tumor-treating fields (TTFields), is a newer treatment for glioblastoma that uses low-intensity energy to disrupt cancer cell division without affecting healthy cells. Delivered via a device resembling a swim cap (Optune), TTFields target glioblastoma cells by interfering with their replication and are typically used for 18 h daily for at least 4 weeks. The most common side effect is skin irritation [31,32].

## 5. Follow-Up

Follow-up care will be carried out by a neurosurgeon and/or a clinical oncologist. And, every 2 to 4 months after the end of radiotherapy, a contrast-enhanced MRI of the brain (or contrast-enhanced CT if there is no MR) will be performed, and then 3 to 6 times a year if there is no relapse or sooner if there is symptomatic progression [7,32].

These scans also monitor brain health and identify any side effects of radiation or chemotherapy. In patients with residual macroscopic disease after surgery, clinical response is assessed using RECIST (Response Evaluation Criteria in Solid Tumors) criteria. If pseudoprogression is suspected, a PET/CT scan will be performed [33].

## 6. Recurrence and Progression

Despite extensive treatment, patients experience tumor recurrence within a median of 6–7 months after treatment, with a high percentage of cases within 2–3 cm of the original lesion margin. High-grade glioma recurrence can arise de novo (primary) or after transformation from a lower-grade glioma (secondary). The accurate diagnosis of tumor recurrence is important because the median overall survival of recurrent HGGs is 4–7 months and because the treatment of pseudoprogression is different from the treatment of tumor recurrence. If the disease has progressed in the brain, patients are considered for salvage treatment (re-operation, second-line chemotherapy, or re-irradiation) on a case-by-case basis (Figure 1) [33,34].

## 7. Discussion

The application of 11C-MET PET imaging in the management of high-grade gliomas represents a significant advancement in the field of neuro-oncology, particularly with respect to enhancing diagnostic accuracy, optimizing treatment planning, and improving post-therapeutic monitoring [14,15]. Our review underscores the pivotal role that 11C-MET PET plays in addressing critical challenges in glioma management, such as the differentiation between tumor recurrence and treatment-related phenomena, including radionecrosis [14,15,16,17].

Globally, approximately 7 out of every 100,000 people are diagnosed with primary central nervous system tumors, with the majority originating from glial cells [6,7]. Gliomas, or tumors of glial origin, are classified into four histological grades, which are crucial for guiding treatment decisions and determining prognosis [7,11,33]. The discovery of several biomarkers for predicting glioma patient prognosis has shown promise. However, despite current treatments, predicting outcomes for high-grade gliomas remains challenging, as relapse often occurs following surgery, radiotherapy, and chemotherapy [6,7,8,9,10,11,12,13,14,15].

For the last four decades, radiotherapy has been the standard treatment following surgical resection for malignant gliomas. However, the optimal target volume for high-grade gliomas remains debated. Current protocols rely on post-operative MRI scans, which often result in relatively large target volumes for radiotherapy planning [7,8,9,10,11,12,13,14,15,16,17,18,19].

Single-photon emission computed tomography (SPECT) imaging of 11C-MET in gliomas enables the visualization of amino acid transport and protein synthesis, although it is less commonly used due to the superior resolution and sensitivity of PET imaging. Integrating PET-CT and MRI into treatment planning for high-grade gliomas offers a detailed assessment of tumor characteristics, enabling clinicians to make more informed and precise treatment decisions [16]. Advanced imaging modalities significantly enhance diagnosis, treatment, and monitoring, improving the quality of life for high-grade glioma patients. Conventional external beam radiotherapy has evolved to include stereotactic and intensity-modulated radiotherapy, both of which have proven effective in improving survival and local control in malignant gliomas [14,15,16,17].

One of the major challenges in the production of 11C-MET for PET imaging lies in the limitations imposed by cyclotron availability and operational stability. The production of 11C-MET depends on the availability of a cyclotron capable of generating the carbon-11 isotope, which has a notably short half-life of approximately 20 min. This brief half-life necessitates on-site production and rapid radiochemical synthesis, as any delay in the process compromises the radiotracer’s activity and diagnostic efficacy. Consequently, the production of 11C-MET is highly sensitive to cyclotron downtime or operational instabilities, which can disrupt the supply chain and limit clinical applications. Additionally, only facilities equipped with a cyclotron and capable of handling the complex logistics of rapid radiotracer synthesis can routinely offer 11C-MET-based imaging, restricting its availability to specialized centers. This infrastructure requirement poses a significant barrier to widespread clinical use, particularly in regions with limited access to cyclotron technology [14,15,16,17,18,19]. The mechanisms governing 11C-MET uptake in gliomas reveal that the tracer’s accumulation is modulated by several key factors, including the overexpression of L-type amino acid transporters, the integrity of the blood–brain barrier, and tumor vascularization. These factors collectively drive the increased metabolic activity observed in high-grade gliomas, which is effectively visualized using 11C-MET PET imaging. The demonstrated correlation between 11C-MET uptake and tumor grade further highlights the tracer’s utility in both accurately delineating tumor extent and assessing malignancy [3,4,5].

11C-MET offers superior specificity in delineating tumor tissue. This amino acid-based tracer is preferentially taken up by glioma cells due to their increased amino acid transport and protein synthesis, resulting in enhanced contrast between tumor and normal brain tissues. The high tumor-to-background ratio afforded by 11C-MET enables a more accurate assessment of tumor extent, aiding in the differentiation between tumor recurrence and radiation-induced necrosis—a common challenge in post-treatment glioblastoma imaging. Additionally, 11C-MET PET has been shown to correlate well with tumor proliferation indices, providing valuable prognostic information and enabling more precise treatment planning. These properties make 11C-MET a valuable tool in the early detection, treatment response evaluation, and management of glioblastoma, contributing to improved patient outcomes [14,15,16,31].

The integration of 11C-MET PET with MRI facilitates a more comprehensive assessment of tumor biology. The combination of anatomical and metabolic data enables the identification of highly active tumor regions, which is crucial for accurate biopsy planning and for tailoring radiotherapy to target the most aggressive components of the tumor. This multimodal approach not only augments diagnostic precision but also holds the potential to improve therapeutic outcomes through the development of more personalized treatment strategies [4,14].

Post-treatment imaging remains one of the most complex aspects of glioma management, particularly in distinguishing between true tumor recurrence and treatment-related effects such as pseudoprogression and radionecrosis [4]. The value of 11C-MET PET in this context is that it is a more reliable method for detecting residual active tumor tissue compared to conventional imaging techniques. The accurate interpretation of these imaging results is critical for avoiding premature alterations in treatment regimens that could adversely affect patient outcomes [14].

In addition to 11C-MET, several other PET imaging radioligands have been explored for glioma imaging, each offering unique advantages and limitations. 18F-fluorodeoxyglucose (18F-FDG) is one of the most widely used PET tracers; however, its utility in glioma imaging is limited due to the high glucose uptake in normal brain tissue, leading to poor tumor contrast. Additionally, inflammatory processes or post-treatment changes, like radiation necrosis, can also exhibit increased 18F-FDG uptake, further complicating the distinction between malignant and non-malignant lesions [35]. A more glioma-specific alternative is 18F-fluoroethyl-tyrosine (18F-FET), an amino acid analog with high specificity for glioma cells, offering improved contrast and lower background uptake in healthy brain tissue [36]. Another promising tracer is 18F-fluorothymidine (18F-FLT), which is a marker of cellular proliferation, providing valuable information about tumor growth dynamics. However, 18F-FLT is limited by lower sensitivity in low-grade gliomas [37]. 18F-DOPA (6-[18F]fluoro-L-DOPA), an analog of L-dopa, has shown efficacy in differentiating recurrent gliomas from radiation necrosis, making it a useful option in assessing treatment response [38]. Each radioligand has its own advantages in terms of tumor specificity, metabolism, and imaging resolution, and the choice of radiotracer depends on the clinical context, the glioma’s biological characteristics, and the specific diagnostic question at hand.

Nevertheless, certain limitations are associated with 11C-MET PET, including the potential for false-positive results in non-neoplastic lesions and the absence of tracer uptake in a subset of WHO grade 2 CNS gliomas with IDH mutations. These findings underscore the need for ongoing research to refine imaging protocols and to develop additional tracers that can address these limitations [39].

Looking ahead, further research is necessary to fully elucidate the mechanisms underlying 11C-MET uptake and to optimize its application in clinical practice. The development of novel tracers and imaging technologies that can enhance or complement the capabilities of 11C-MET PET may play a crucial role in advancing the field. Additionally, large-scale clinical trials are essential to validate the findings of this study and to establish standardized guidelines for the use of 11C-MET PET in glioma management [31,32,33,39].

Recent studies have expanded the potential applications of 11C-MET-PET beyond its traditional role in cancer imaging, particularly in glioma diagnostics. Emerging evidence suggests that MET-PET can be useful in non-cancer applications, such as the evaluation of inflammatory and neurodegenerative conditions [40]. For instance, MET-PET has shown promise in identifying active regions of inflammation in diseases like sarcoidosis, where amino acid uptake correlates with inflammatory cell activity [41]. Additionally, MET-PET is being investigated for its potential to visualize protein synthesis and metabolic activity in neurodegenerative diseases, including Alzheimer’s disease and multiple sclerosis [42]. These novel applications of MET-PET in detecting non-cancerous pathological processes highlight its versatility and potential as a diagnostic tool in broader clinical settings.

## 8. Conclusions

In conclusion, high-grade gliomas, particularly glioblastoma multiforme, remain a significant clinical challenge due to their aggressive nature, high recurrence rates, and limited treatment advancements over the past decade. The integration of advanced imaging modalities, such as PET-CT and MRI, into treatment planning has enhanced the precision of diagnosis and therapeutic strategies, contributing to improved patient outcomes. Among these, 11C-MET PET stands out for its superior specificity in tumor delineation and its ability to differentiate between tumor recurrence and treatment-induced changes. Despite the logistical challenges associated with its short half-life, 11C-MET offers invaluable insights into tumor biology, aiding in early detection, treatment response evaluation, and overall management of glioblastoma, ultimately improving the prognosis for affected patients.

## Figures and Tables

**Figure 1 cancers-16-03200-f001:**
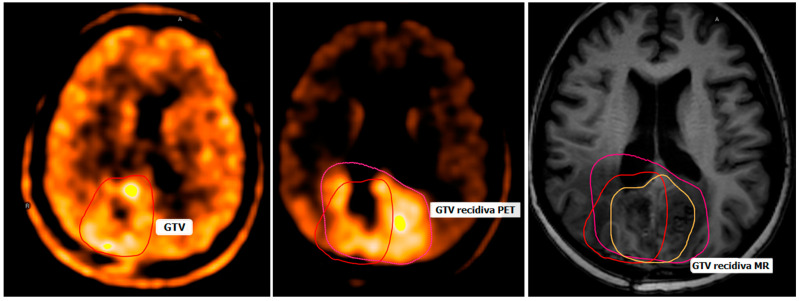
11C-MET fusion PET-CT and post-contrast T1-weighted MR axial brain images. It can be seen that the recurrence of the GTV highlighted by MET-PET (pink) is significantly larger than that highlighted by MR (orange). The red outline in the image indicates the primary GTV.

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
