# Peer review of "Value of 11C-Methionine PET Imaging in High-Grade Gliomas: A Narrative Review"

_cancers, 2024, doi:10.3390/cancers16183200_

Round 1
Reviewer 1 Report
Comments and Suggestions for Authors
This is an appropriate review to call attention to the clinical benefits of MET-PET imaging for neuro-oncologists regarding glioma patients. As this diagnostic tool combined with radiotherapy facilitates personalized oncotherapy, the message of the manuscript makes it valuable enough for publication.
Author Response
Dear Reviewer 1!
Thank you sincerely for your thoughtful and constructive feedback on our work. We appreciate your time and expertise.
Regarding your comments:
This is an appropriate review to call attention to the clinical benefits of MET-PET imaging for neuro-oncologists regarding glioma patients. As this diagnostic tool combined with radiotherapy facilitates personalized oncotherapy, the message of the manuscript makes it valuable enough for publication. - THANK YOU VERY MUCH FOR YOUR POSITIVE FEEDBACK
Reviewer 2 Report
Comments and Suggestions for Authors
The authors did an excellent job compiling a nice review article on the merits of [11C]methionine PET for tumor imaging applications. The article is well-written for the most part, with a good flow, and ease to follow.
minor edits:
1. double check the spellings including tumors and use one tense (past tense is good--since it is a review)
2. highlight the drawbacks of cyclotron availability and stability
3. exclusive discussion on other options i.e., other PET imaging radioligands
4. A few lines and new applications of MET-PET for non-cancer applications
5. image analyses and outcome profile
Author Response
Dear Reviewer 2!
We extend our sincere gratitude for your thorough review of our work. Your constructive feedback and valuable suggestions have significantly enhanced the quality of our research. We appreciate your time, expertise, and thoughtful contributions to the refinement of our manuscript.
Regarding your comments:
The authors did an excellent job compiling a nice review article on the merits of [11C]methionine PET for tumor imaging applications. The article is well-written for the most part, with a good flow, and ease to follow.
minor edits:
- double check the spellings including tumors and use one tense (past tense is good--since it is a review)- THANK YOU FOR YOUR FEEDBACK, WE’VE CORRECTED
- highlight the drawbacks of cyclotron availability and stability- THANK YOU, WE’VE ADDED A SEGMENT WHICH IS HIGHLIGTING THE MENTIONED ASPECTS
- exclusive discussion on other options i.e., other PET imaging radioligands - THANK YOU, WE’VE MENTIONED FEW RADIOGLANDS REGARDING GLIOMA IMAGING
- A few lines and new applications of MET-PET for non-cancer applications- THANK YOU, WE’VE ADDED FEW LINES REGARDING NON-CANCER APPLICATIONS
- image analyses and outcome profile- WE’VE EXPLAINED THE IMAGE ANALYSES AND OUTCOME PROFILE
Reviewer 3 Report
Comments and Suggestions for Authors
This manuscript describes the general management of glioma. Other imaging modalities, such as SPECT or FDG PET should be included in this discussion.
The value and limitation of FDG PET were described without exact research results. Author should provide clinical index such as false positive rates.
All abbreviations should be fully spelled out at first appearance and should be used after the definitions.
Author Response
Dear Reviewer 3,
Thank you immensely for your insightful and detailed review of our manuscript. Your thoughtful comments and suggestions have proven invaluable in strengthening the overall quality of our work. We are genuinely appreciative of your time and expertise in contributing to the refinement of our research.
Regarding your comments:
This manuscript describes the general management of glioma. Other imaging modalities, such as SPECT or FDG PET should be included in this discussion. - THANK YOU FOR YOUR COMMENT, WE’VE MENTIONED SPECT AND FDG PET IMAGING DURING DISCUSSION
The value and limitation of FDG PET were described without exact research results. Author should provide clinical index such as false positive rates. - THANK YOU, WE’VE ADDED SOME THOUGHTS REGARDING 18F-FDG IMAGING
All abbreviations should be fully spelled out at first appearance and should be used after the definitions. - WE’VE TRIED TO REDEFINE
Round 2
Reviewer 3 Report
Comments and Suggestions for Authors
I found significant improvements in this manuscript, so this is acceptable now.